# Reinforcement Effect of a Concrete Mat to Prevent Ground Collapses Due to Buried Pipe Damage

**Jeongjun Park [1]** , **Yoonseok Chung [2,\*]** **and Gigwon Hong [3,\*]**

[1]   Incheon Disaster Prevention Research Center, Incheon National University, Incheon 22012, Korea; smearjun@hanmail.net

[2]   International Business Division, Korea Conformity Laboratories, Seoul 08503, Korea

[3]   Institute of Technology Research and Development, Korea Engineering & Construction, Seoul 05661, Korea

\*   Correspondence: yschung24@kcl.re.kr (Y.C.); gigwon_hong@kecgroup.kr (G.H.)

**Abstract:** This study described a ground reinforcement effect of a concrete mat, in order to apply a concrete mat for ground subsidence restoration of an open cut. A concrete mat can prevent the expansion of a cavity and relaxation area underground due to buried pipe damage when the buried pipe is in use. An experimental study was conducted to analyze the stress distribution characteristics of an underground area by ground reinforcement of a concrete mat. In addition, a numerical analysis was performed to estimate the range of underground reinforcement of a concrete mat. As an experiment results, the maximum stress reduction ratio of the concrete mat in the underground was 28.5% to 30.9%, which means the reinforcement effect of the concrete mat, according to the installation depth of the concrete mat. The finite element analysis (FEA) results showed that the installation depth of the concrete mat differed in various scenarios, in order to secure the reinforcement effect of the concrete mat according to the load conditions (point and uniform load). Therefore, the reinforced depth of a concrete mat should be determined by the load type on the surface.

**Keywords:** concrete mat; ground reinforcement; ground subsidence; finite element analysis

## 1. Introduction

Recently, there have been several occurrences of ground subsidence in major cities in Korea. One of the major causes of ground subsidence has been attributed to underground cavities. Underground cavities are generated by soil loss due to sewage damage, poor compaction of the soil around pipes, and inadequate measures for underground water because of excavations [1–3]. In particular, according to a study by the Korea Institute of Geoscience and Mineral Resources [4], approximately 85% of 3200 cases of ground subsidence that occurred in Seoul from 2010 to the first half of 2015 can be attributed to sewage damage. Bae, et al. [5] conducted research on underground cavities, which are the major cause of ground subsidence. They divided the cavities into area with damaged sewage and area near an excavation. From this, they determined that approximately 82% of the cases were caused by damaged sewerage. Most of the ground subsidence cases in urban areas in Korea occur as the surrounding soil flows into the damaged sewage, thus generating an underground cavity. Therefore, technology for minimizing sewage damage is needed to prevent the occurrence of ground subsidence.

The mechanism of generating underground cavities due to damaged sewage must be investigated in order to prevent the sewage damage [6]. Several studies have been conducted in Korea using experiments and numerical analyses to examine the mechanism of generating underground cavities. Kim and Umm [7] analyzed the effects of discharged underground soil that had been disturbed by the flow of underground water during the expansion of underground cavities. Additionally, Lee, et al. [8] used a discrete element method to investigate how the settlement of the ground surface increased as the

relaxation zone of the surrounding soil and the cavity generated due to the damaged sewage expanded. Moreover, Kim, et al. [9] simulated the relaxation zone of the surrounding soil and the underground cavity using a discrete element method based on the results of a laboratory model test. Furthermore, Lee, et al. [10] evaluated the applicability of a numerical simulation for the phenomenon of ground subsidence based on a numerical analysis using a large displacement analysis method, while analyzing the effects of a reduction in the unsaturated soil strength due to a rise in the groundwater level. You, et al. [6] identified the distribution characteristics of the void ratio of the soil around a cavity due to soil loss based on finite element analysis. Additionally, You, et al. also examined how to determine the boundary of the relaxation zone by analyzing the distribution characteristics of the shear stress reduction ratio of soil around a cavity. Moreover, Lee, et al. [11] inspected the correlation according to the load condition that was applied to the asphalt pavement layer by considering the thickness of the asphalt pavement layer, the soil depth, and the width and height of the cavity as influential factors. These factors are the criteria employed for the underground cavity grade system in Korea (Seoul). Jeong, et al. [12] experimentally evaluated the relaxation of the ground and the scale of an underground cavity according to the mixing ratio of sand and clay. Furthermore, Jeong, et al. [12] discovered that the scale of ground subsidence can be reduced when the clay content is decreased. In addition to this, Cooper [13] analyzed the ground subsidence phenomenon that occurred over an extended period of time. They [13] reported that ground subsidence was caused by an underground cavity that was found in a gypsum layer. A disaster prediction map was produced for areas with a high risk of ground subsidence. Tharp [14] conducted a study on the mechanism of sinkholes and suggested that abrupt changes in the pore pressure around an underground cavity affect the occurrence of sinkholes. Moreover, Drumm, et al. [15] proposed a stability chart based on a stability evaluation of ground subsidence in areas where karst is formed. They [15] did this while performing a two-dimensional finite element method (FEM) numerical analysis in which the shear stress reduction method was applied. From this, they [15] suggested a safety factor calculation method for ground where cavities are found. Additionally, Suchowerska, et al. [16] analyzed the correlation between ground settlement and destruction in the upper part of a cavity due to an underground cavity by applying the ground condition, cavity shape, and presence of an underground anomaly zone as influential factors. Furthermore, Kuwano, et al. [17] analyzed the effect of the underground water level such that it generates cavities, and this was achieved by performing a laboratory model test. From these results, it is known that a relaxation zone occurs in the area around underground cavities.

A relaxation zone refers to ground that is in a stress release state or a relaxed state due to a reduction in the rigidity or compactness around the excavation area when the excavation is performed. This must be taken into consideration for ensuring the stability of the excavated ground. Specifically, the size of underground cavity may expand as the area of the relaxation zone increases. Therefore, the expansion of the cavity can be prevented if the expansion of a relaxation zone is avoided after a cavity is generated due to damage of the buried pipes.

Meanwhile, the restoration method of underground cavity that causes ground subsidence can be divided into an open cut method and a trenchless method. One of the examples of the trenchless method of restoring underground cavities is the grouting method, which has the advantage of filling the voids in the ground; however, it causes environmental pollution. Specifically, a cement-based filler and a liquid solution are used together, and the applied area cannot be controlled depending on the injection method. Furthermore, there is a possibility of leakage of heavy metals. An open-cut method for restoring the ground subsidence typically involves controlling the traffic around the road where the underground cavity has occurred, excavating the road and filling the cavity, compacting the soil, and then reinstalling the road structures. This method may induce strength degradation of the ground due to a disturbance in the ground from the excavation, segregation, or poor compaction. The restored soil may be lost again, which may regenerate the underground cavity. Despite these drawbacks, using an open cut method is inevitable in areas where ground subsidence has occurred when replacing the pipe lines and the group collapses due to the deteriorated buried facilities or damaged pipe lines.

In this study, the reinforcement effect was evaluated by using a concrete mat that was developed to prevent ground subsidence by suppressing the expansion of the relaxation zone and cavity, which may occur while buried pipes are in service. This was performed when an open cut method was applied to the ground subsidence that had occurred in the area with buried pipes. Specifically, a laboratory test was conducted to analyze the characteristics of the stress occurrence according to the concrete mat reinforcement. Moreover, a numerical analysis was performed to estimate the range of underground reinforcement of a concrete mat that is installed on top of the buried pipes. The results were analyzed to examine the stress reduction in the ground.

## 2. Ground Reinforcement Method

### 2.1. Literature Review

Protecting buried pipes requires technology for minimizing the deformation of the ground near the buried pipes. Several studies have been conducted on ground reinforcement methods to prevent ground subsidence and protect buried pipes. Corey, et al. [18] analyzed the relationship between the longitudinal deformation of buried pipes and the static load by forming a geosynthetic layer on top of the buried pipes. In order to achieve this, they [18] conducted an experimental study on the sagging and deformation characteristics of the buried pipes against a static load. Elshimi and Moore [19] assessed the earth pressure that acts on buried pipes according to the distribution of the bearing pressure and the ground conditions. This was done while considering that the underground movement of buried pipes is significantly affected by the characteristics of the backfilling materials. Tafreshi and Khalaj [20] conducted an experiment on the deformation of a high-density polyethylene (HDPE) pipe in reinforced sand under a repeated load. They [20] also analyzed the behavioral relationship according to the relative density of the soil and the pipe burial depth. Tahmasebipoor, et al. [21] simulated geosynthetic reinforced ground in the upper part of the cavity. They carried out a numerical analysis according to the tensile strength of the geosynthetic, the length of the reinforcement layer, and the cavity size, for examining the correlation between the size of the cavity and the ground reinforcement conditions. According to a study by Palmeira and Andrade [22], the stress and deformation generated near the buried pipes were reduced when a geosynthetic layer was installed near the buried pipes. Hegde, et al. [23] and Hegde and Sitharam [24] conducted experiments and numerical analyses for reinforcing the sand with geocells. They discovered that the geocells distribute the underground stress in the lateral direction, thereby reducing the load that is acting on the buried pipes. Tafreshi, et al. [25] conducted a plate bearing test by reinforcing a foundation structure and the lower part of the road pavement with several layers of geocells. They [25] discovered that the underground stress was reduced as the number of geocell layers and the elastic modulus of the ground increased. Pancar and AkpJnar [26] studied the ground reinforcement effect using a variety of geosynthetic materials to improve the bearing capacity of the ground under the pavements. Additionally, Naggar, et al. [27] confirmed, through experimental and numerical analysis, that a geogrid bridging platform can decrease the excessive stress and the deformation generated in buried pipes. Mehrjardi, et al. [28] also conducted a numerical analysis on an experimental model for protecting buried pipes, while considering the sand-rubber mixed soil and the geocell-reinforced foundation. Accordingly, when the geocell and sand-rubber mixed soil are used simultaneously, the ground surface settlement and the sagging of the buried pipes are reduced, even when a repeated load is applied to the ground surface. Ali and Choi [29] examined the risk index while considering a variety of factors that influence the occurrence of ground subsidence, such as damage in the water supply system in cities. Other experimental and numerical analyses on ground reinforcement technology for protecting buried pipes and on technology for reducing underground stress using reinforcement materials have been performed.

Most previous studies used geosynthetic materials for ground reinforcement to protect buried pipes; however, the flexibility of geosynthetic materials causes the sagging of materials under a

continuous load. As a result, this limits the prevention of underground cavity expansion and the relaxation zone. Therefore, this study examined the effect of underground stress reduction by using a concrete mat with a high strength as a reinforcement material.

### 2.2. Concept of Ground Reinforcement

A concrete mat is used as a stiffener to resist the overburdened load by reinforcing the area where strength degradation may occur in the ground due to poor compaction. The length and width of a concrete mat can be easily adjusted; hence, it can be conveniently applied according to the restoration area conditions. This includes an area where soil loss has occurred due to buried pipe damage or an area where an underground cavity has occurred (Figure 1).

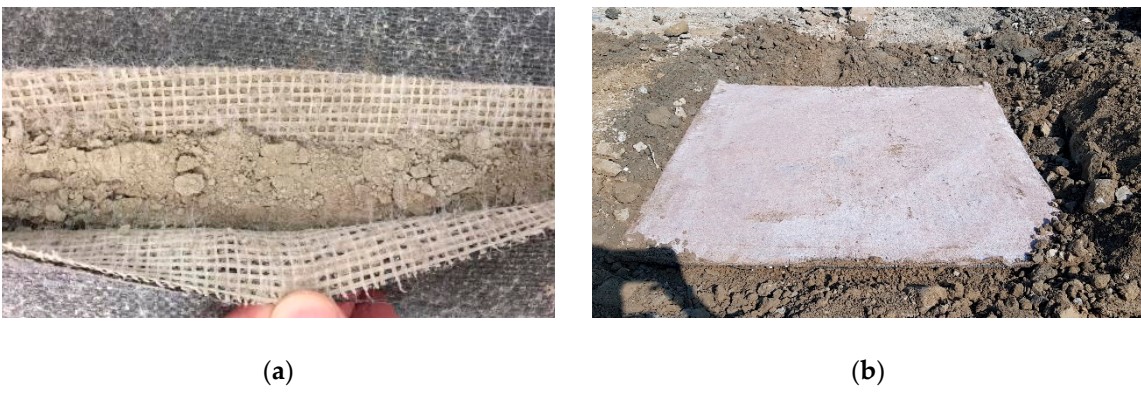

(**a**) (**b**)

**Figure 1.** Concrete mat: (**a**) composition of a concrete mat; (**b**) field application example.

Figure 2 shows a schematic of the open cut method for ground subsidence restoration using a concrete mat and the relevant principle. Figure 2a illustrates the process of ground subsidence occurrence as a load is applied to the ground surface when the buried pipe damage leads to the occurrence and expansion of the underground cavity. Figure 2b,c shows the typical restoration method involving backfilling the soil and the open cut method employed for restoring the underground cavity using a concrete mat, respectively. Figure 2d illustrates the effect of a concrete mat when an underground cavity occurs. Accordingly, the characteristics of the restoration method employed for ground subsidence using a concrete mat are as follows.

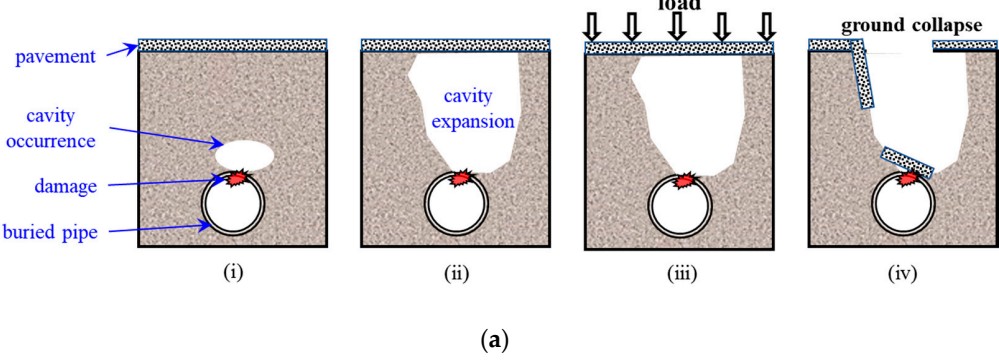

(**a**)

**Figure 2.** *Cont.*

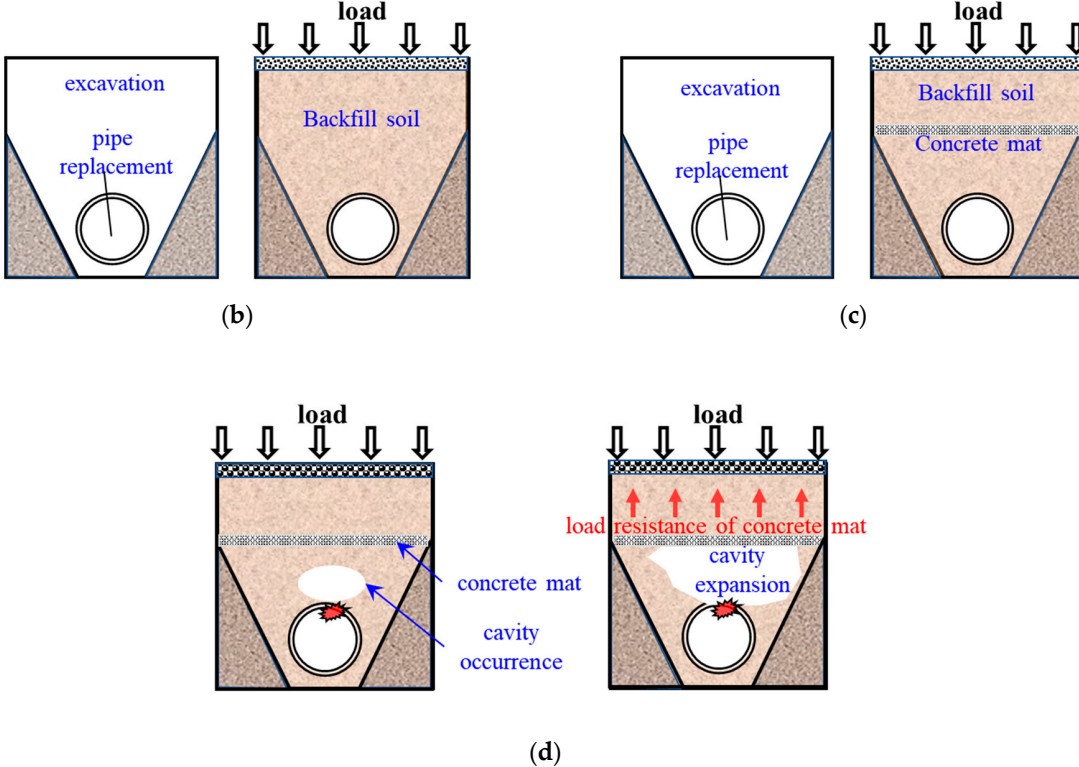

**Figure 2.** Schematic of the ground subsidence restoration and the ground reinforcement using a concrete mat: (**a**) process of ground subsidence occurrence due to an underground cavity; (**b**) typical restoration method employed for ground subsidence occurrence; (**c**) restoration method using a concrete mat for ground subsidence occurrence; (**d**) role of the concrete mat in underground cavity occurrence.

When it is reinforced with backfilled soil, an underground cavity expands, which may lead to ground subsidence at the ground surface when the load is repeatedly applied. However, when reinforced with a concrete mat, a cavity that occurs around the pipes cannot expand beyond the concrete mat. In other words, the relaxation zone in the underground cavity cannot expand. As a result, sudden ground subsidence can be prevented by the stiffness of a concrete mat, even when the load is applied to the ground surface. The reinforcement effect of a concrete mat can minimize the damage when the underground cavity occurs again, while maintaining the restored cavity.

## 3. Experimental Investigation

As previously explained, the open cut method employed for restoring ground subsidence involves excavating the cavity area, filling the cavity with soil, and then reinstalling the structures on top. Despite the drawbacks of material segregation, disturbance in the surrounding soil, and poor compaction, this method must be applied when ground subsidence occurs due to damage of the buried pipes.

The open cut method uses a concrete mat that can prevent the ground surface from suddenly collapsing due to the resistance against the top load based on the strength of the material. Therefore, a plate bearing test was performed in a laboratory to quantitatively analyze the reinforcement effect of a concrete mat based on the characteristics in which the stress occurrence was analyzed, according to whether the concrete mat was reinforced.

### 3.1. Materials

Jumunjin sand, which is the standard sand in Korea, has a uniform particle size and was used in a soil sample in the experiment to ensure that the stress was evenly distributed over the load on the surface. Table 1 lists the engineering properties of the Jumunjin sand. The Atterberg limit test revealed

that the soil sample did not have plasticity, and it was classified as poorly graded sand (SP), according to the Unified Soil Classification System (USCS).

**Table 1.** Engineering properties of sand.

| Specific Gravity | Sieve Analysis | | Compaction Test | | USCS |
|---|---|---|---|---|---|
| | $C_u$ | $C_g$ | $\gamma_{d(max)}$ (KN/m³) | $\omega_{opt}$ (%) | |
| 2.61 | 3.75 | 0.98 | 16.2 | 7.8 | SP |

A concrete mat consists of geosynthetic and fill materials (e.g., cement and fine aggregates). Specifically, cement and fine aggregates were applied to fill the space between the top and bottom geosynthetic materials, and watering was performed to secure a sufficient strength for the experiment. A concrete mat is not expandable and has a high strength. The strength of the concrete between geosynthetics was 40 MPa, and rapid hardening cement was applied for ground reinforcement.

*3.2. Experiment Details*

Figure 3 shows a schematic diagram of the plate bearing test that was conducted in the laboratory to verify underground stress occurrence according to the reinforcement of a concrete mat.

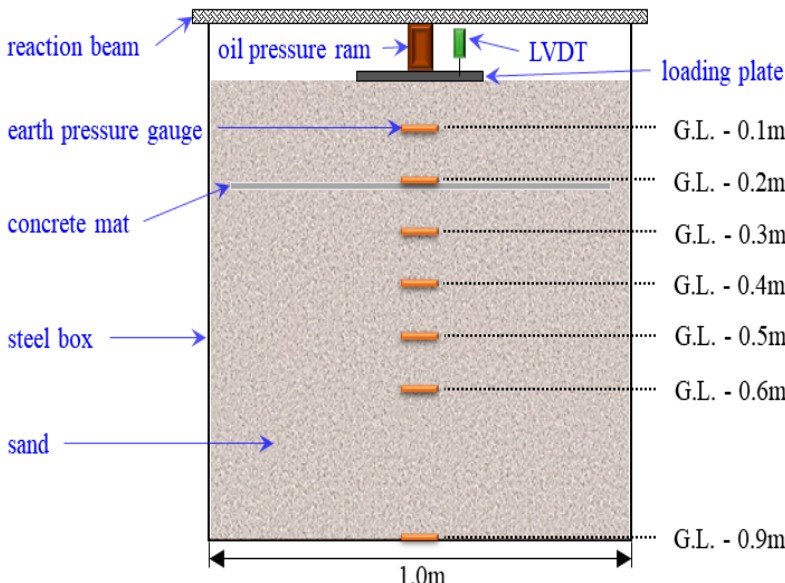

**Figure 3.** Schematic of the plate bearing test on a laboratory scale.

The ground sample was formed using standard sand in a steel box (1.0 (L) × 1.0 (B) × 0.9 (H)) with a medium density condition. The earth pressure gauge for checking the stress occurrence was installed at G.L. (Ground Level) −0.1 m to G.L. −0.6 m (at 0.1 m intervals) from the center of the steel box in the vertical direction and at the very bottom (G.L. −0.9 m). The diameter of the earth pressure gauge was 0.05 m, and the capacity could be measured up to 500 kPa.

Linear variable differential transformers (LVDTs, 2ea) were installed on the loading plate to measure the settlement of the loading plate (ground surface settlement). Earth pressure gauges (7ea) were applied to measure the stress generated underground, as shown in Figure 3. In addition, the load transferred to the underground area by the loading plate caused maximum stress in the vertical direction. Therefore, in order to measure the maximum stress due to the vertical load affecting the buried pipe damage under the plane stress condition, an earth pressure gauge was installed in the center of the steel box.

Three types of experiments were conducted, in which the concrete mat was installed at G.L. −0.1 m and G.L. −0.2 m from the ground surface and the concrete mat was not installed. The size of the concrete mat applied in this study was 0.9 m × 0.9 m (W × L). Lubrication was performed using oil and plastic wrap to minimize the friction between the soil and wall face of the steel box when the load was transferred underground. The load was applied sequentially from 4, 8, and 12 kPa, in order to minimize the reduction in stress due to soil failure.

The experimental procedure is shown in Figure 4a presents an overall view of the ground sample and the earth pressure gauge installed at a planned depth. Figure 4b shows the concrete mat installed at a planned depth. After the concrete mat and the earth pressure gauge were installed and the ground sample was completed, a loading plate was installed for applying a load. A leveler was used to prevent eccentricity during plate loading (Figure 4c). As shown in Figure 4d, a reaction beam and an oil pressure ram were mounted. In addition, a displacement gauge—linear variable differential transformer (LVDT)—was installed for measuring ground settlement due to the load (Figure 4e).

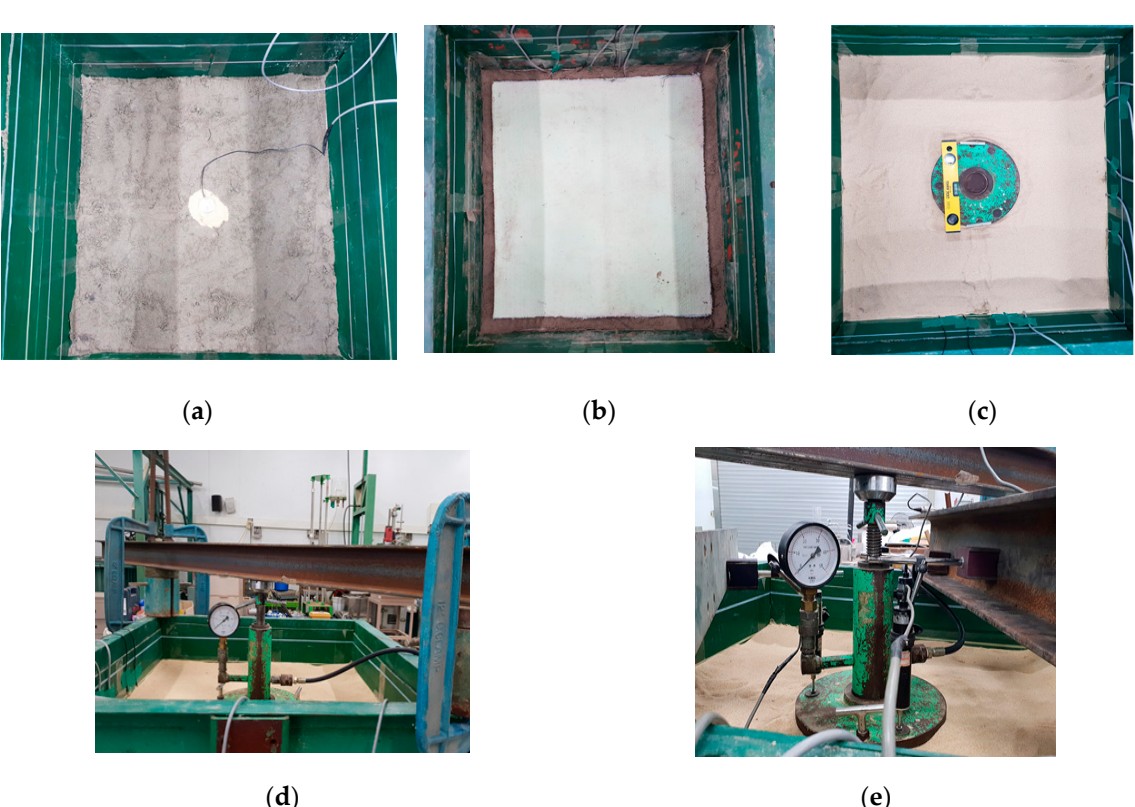

(**a**)　　　　　　　　　　　　　　　　(**b**)　　　　　　　　　　　　　　　　(**c**)

(**d**)　　　　　　　　　　　　　　　　　　　　　　　　(**e**)

**Figure 4.** Experimental procedure: (**a**) earth pressure gauge installation in soil; (**b**) concrete mat installation; (**c**) loading plate installation; (**d**) reaction beam installation; (**e**) linear variable differential transformer (LVDT) installation.

### 3.3. Results

Figure 5 shows the stress occurrence per depth according to the installation depth of concrete mat. Figure 5a shows the result obtained when the concrete mat was not applied. As the depth increased, the stress occurrence decreased nonlinearly under all load conditions. Moreover, as the stress level increased from 4 to 12 kPa, the reduction in stress occurrence increased with the depth. Figure 5b,c illustrate the experimental results according to the reinforced depth of the concrete mat at 0.1 and 0.2 m from the surface, respectively. When compared to the non-reinforced case, the stress occurrence was significantly reduced from the reinforced depth or below. Specifically, the stress-reducing effect occurring below the buried depth of the concrete mat appears to be due to the stiffness of the concrete mat.

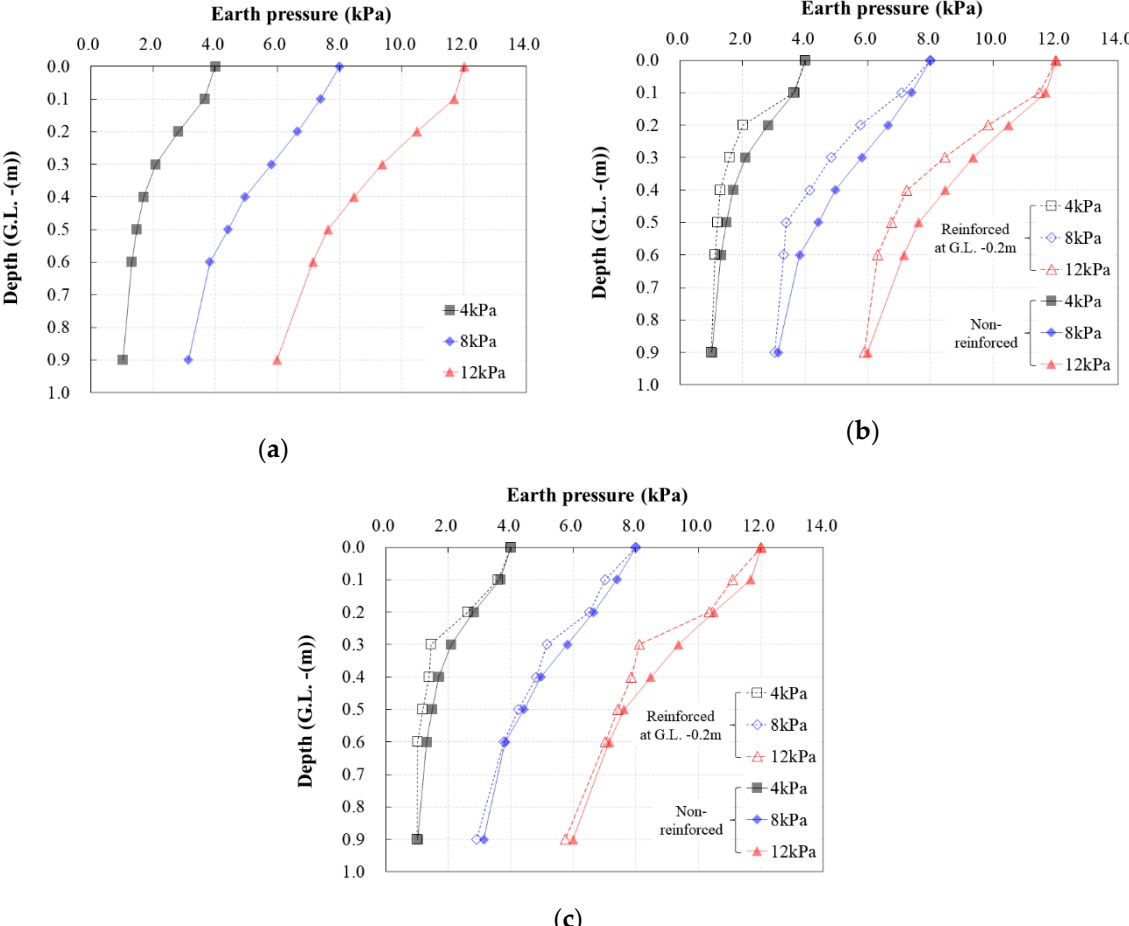

**Figure 5.** The relationship between the earth pressure and depth, according to the load step: (**a**) non-reinforced; (**b**) reinforced at G.L. −0.1 m; (**c**) reinforced at G.L. −0.2 m.

Using the experimental results, the stress reduction ratio was calculated per depth, according to the reinforcement of the concrete mat, as shown in Equation (1). Table 2 lists the calculation results.

$$\text{Stress reduction ratio}(\%) = (P_{(non)} - P_{(rc)})/(P_{(non)}) \tag{1}$$

where $P_{(non)}$ represents the stress occurrence per depth when not reinforced with the concrete mat and $P_{(rc)}$ represents the stress occurrence per depth when reinforced with the concrete mat.

**Table 2.** Comparison of the reinforced ground and non-reinforced ground in terms of the stress reduction ratio.

| Depth (m) | Reinforced at G.L. −0.1 m | | | Reinforced at G.L. −0.2 m | | |
|:---:|:---:|:---:|:---:|:---:|:---:|:---:|
| | 4 kPa | 8 kPa | 12 kPa | 4 kPa | 8 kPa | 12 kPa |
| 0 | 0.0 | 0.0 | 0.0 | 0.0 | 0.0 | 0.0 |
| 0.1 | 0.7 | 4.3 | 1.8 | 1.8 | 4.8 | 5.0 |
| 0.2 | 28.5 | 13.4 | 6.2 | 6.6 | 2.1 | 1.4 |
| 0.3 | 23.8 | 16.9 | 9.7 | 30.9 | 11.5 | 13.4 |
| 0.4 | 24.5 | 16.5 | 14.5 | 18.9 | 3.2 | 7.3 |
| 0.5 | 18.7 | 23.2 | 11.4 | 20.0 | 4.1 | 2.6 |
| 0.6 | 13.7 | 13.7 | 11.6 | 22.4 | 1.2 | 1.8 |
| 0.9 | 3.1 | 3.1 | 1.8 | 2.4 | 7.5 | 4.3 |

When the concrete mat was installed at G.L. −0.1 m, the stress reduction continued below a 0.2 m depth, while the stress reduction ratio decreased. Moreover, the stress reduction ratio decreased as the applied load increased. When the concrete mat was installed at G.L. −0.2 m, the stress occurrence also decreased below the burial depth of the concrete mat. Despite the inconsistency, the stress reduction ratio also decreased as the applied load increased. Based on these results, a concrete mat reduces the stress by up to 28.5–30.9%, depending on the burial depth and the condition of the pressure applied to the ground surface.

## 4. Numerical Analysis

### 4.1. Finite Element Analysis (FEA)

A two-dimensional finite element analysis (FEA) was performed using ABAQUS, which is useful for evaluating the soil–structure relationship and analyzing the stress reduction effect induced through the stress occurrence of the concrete mat for various load conditions. The FEA was verified using the results of the plate loading test for the non-reinforced ground. Using the verified model, the stress reduction effect was analyzed in a consideration of the density of the ground, the type of load, and the application conditions of the concrete mat.

The engineering properties of the soil and concrete mat used in the analysis are shown in Table 3. First, the unit weight and internal friction angle of the sand were applied with the experimental research. The elastic modulus and Poisson's ratio of sand were obtained from the results of a study by Hunt [30]. Hunt's research on the engineering properties of sand is one of the most frequently referenced studies in Korea. The elastic modulus, density, and Poisson's ratio of the concrete mat considering the concrete strength were obtained from previous studies [31,32].

**Table 3.** Properties of the soil and concrete mat.

| Classification | Soil (Sand) | | | Concrete Mat |
|---|---|---|---|---|
| **Material Model** | **Elastic and Plastic (Mohr–Coulomb)** | | | **Elastic** |
| Elastic modulus (kPa) | Density condition of sand | | | 27,800 |
| | Loose | Medium | Dense | |
| | 785 | 1570 | 2945 | |
| Unit weight (kN/m$^3$) | 16.2 | | | 24.5 |
| Poisson's ratio | 0.25 | | | 0.2 |
| Friction angle (°) | 32.8 | | | - |
| Cohesion (kPa) | 0 | | | - |

To verify the analysis model, modeling was conducted, in which a plate bearing test was taken into consideration, as shown in Figure 6. In order to develop a reasonable analysis model, the analysis of the stress change in the ground according to the load range, ground width, and depth was performed in the case of the non-reinforced ground. Here, 4 kPa of uniform load was applied, and the analysis was conducted, in which the concrete mat was buried at 0.1 m for verifying the analysis results of applying the concrete mat.

The analysis results for the uniform load range, as well as the width and depth of the ground, are presented in Figure 7 for the condition where the concrete mat was not applied. As Figure 7a shows, when the uniform load range in the FEA was large (uniform load range = 0.3 m), the analysis and experimental results varied as the soil depth increased; however, the tendency of the underground stress was similar for both results. Based on these results, when the analysis was performed for the different soil widths and depths while fixing the pressure range at 0.3 m, no significant difference was

observed. As the diameter of the loading plate was 0.3 m during the plate bearing test, the analysis results for applying the same pressure range were determined to be reasonable.

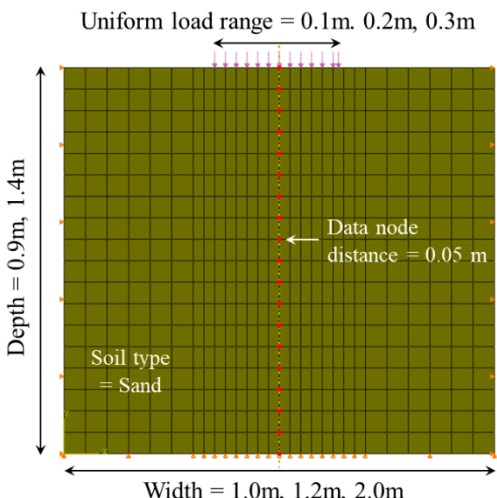

**Figure 6.** Finite element analysis (FEA) model produced for verification.

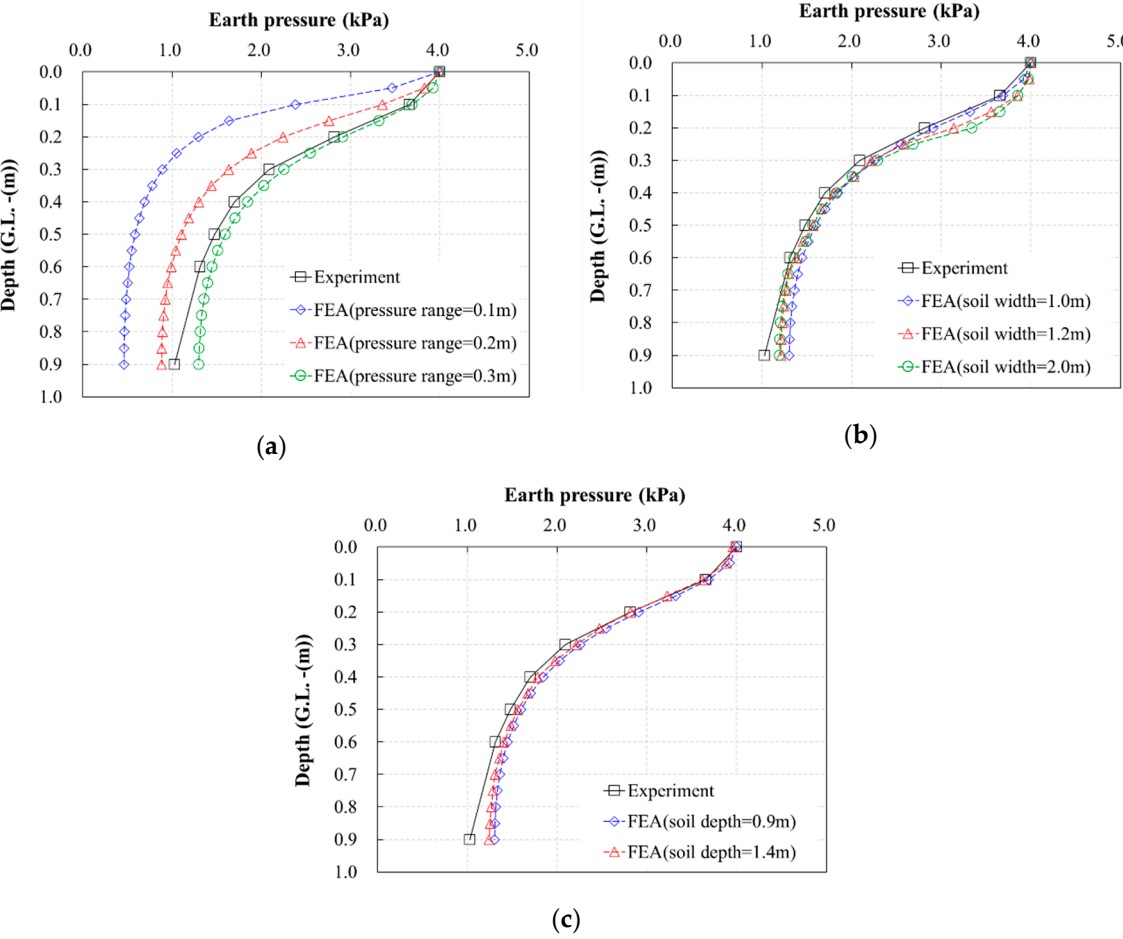

**Figure 7.** Analysis results for model verification: (**a**) effect of the pressure range; (**b**) effect of the soil length; (**c**) effect of the soil depth.

Using the analysis model that was previously mentioned, an analysis was performed for when the concrete mat was installed at G.L. −0.1 m and G.L. −0.2 m. Figure 8 presents a comparison

of the experimental and FEA results. First, as mentioned in Section 3.3, the stress reduction effect was approximately 28.5–30.9% at the depth directly below where the soil was reinforced with the concrete mat when 4 kPa of load was applied in the laboratory test. When the stress reduction effect was analyzed according to the reinforcement concrete mat using the numerical analysis results, the stress reduction effect was approximately 22–25% when the soil was reinforced with the concrete mat. Accordingly, the difference from the experimental results is assumed to have been caused by the interface model. This interface model simulated the behavior between the concrete mat and the ground; that is, it was assumed that the interface model was ignored to secure the simplicity and usability of the numerical analysis model in this study. However, when considering the limitation of the numerical simulation, the similarity in the stress reduction tendency per depth in the experiment and the analysis results, as well as the difference in the stress reduction effect, did not significantly influence the simulation of the stress behavior. Therefore, based on the results of comparing the analysis results and the experimental results for the verification model, it was determined that further numerical analysis studies using the analysis model would be possible.

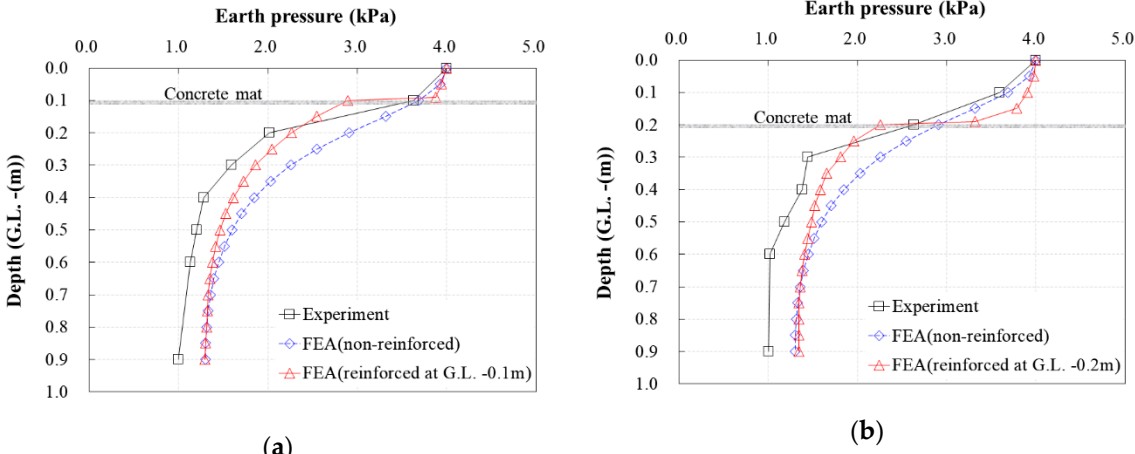

**Figure 8.** Comparison of the experimental and FEA results for the reinforced concrete mat: (**a**) case of reinforcement at G.L. −0.1 m; (**b**); case of reinforcement at G.L. −0.2 m.

*4.2. FEA Based on the Condition of the Soil Density and the Load Type*

Using the verified model, the stress reduction effect was analyzed in a consideration of the density of the ground, the type of load, and the application conditions of the concrete mat. Using the research results, the FEA was performed using the verified model, in order to apply the various soil densities and load types, depending on whether or not a concrete mat was applied. Here, the relative density was considered for the soil density, while the point load and uniform load were applied for the load type. Moreover, the size of the concrete mat was assumed to be 1 m while considering the diameter of the buried pipes (D = 0.3 m–0.5 m) and the scale of the excavation work. The burial depth of the concrete mat was G.L. −0.1 m-G.L. −0.3 m. The material properties required for the analysis are listed in Table 3, and the analysis model is illustrated in Figure 9.

For the analysis, a load was applied to the ground surface until damage occurred in the center of the concrete mat, and the correlation with the displacement was then analyzed.

Figure 10 shows the relationship between the failure load and the displacement of the concrete mat, according to the soil density and the reinforced depth for each load type.

As demonstrated in Figure 10a, the load during the failure of the concrete mat increased as the relative density increased for the case of a point load. Regardless of the relative density condition, similar failure loads were observed while excluding the concrete mat that was buried at G.L. −0.1 m. The failure load was generated with approximately a 4.5–5.5% displacement of the length of the concrete mat. In contrast, when the concrete mat was buried at G.L. −0.1 m, the failure load increased

by approximately 60–110%, according to each relative density. The displacement was around 7.5–8.0%. The deformation rate lessened after the failure load in all analysis results. This is because failure occurred in the concrete mat, but not in the soil.

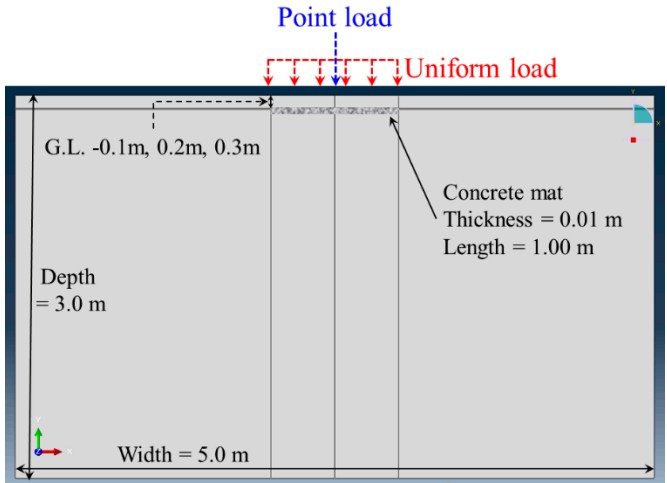

**Figure 9.** FEA model considering the soil density and the load type.

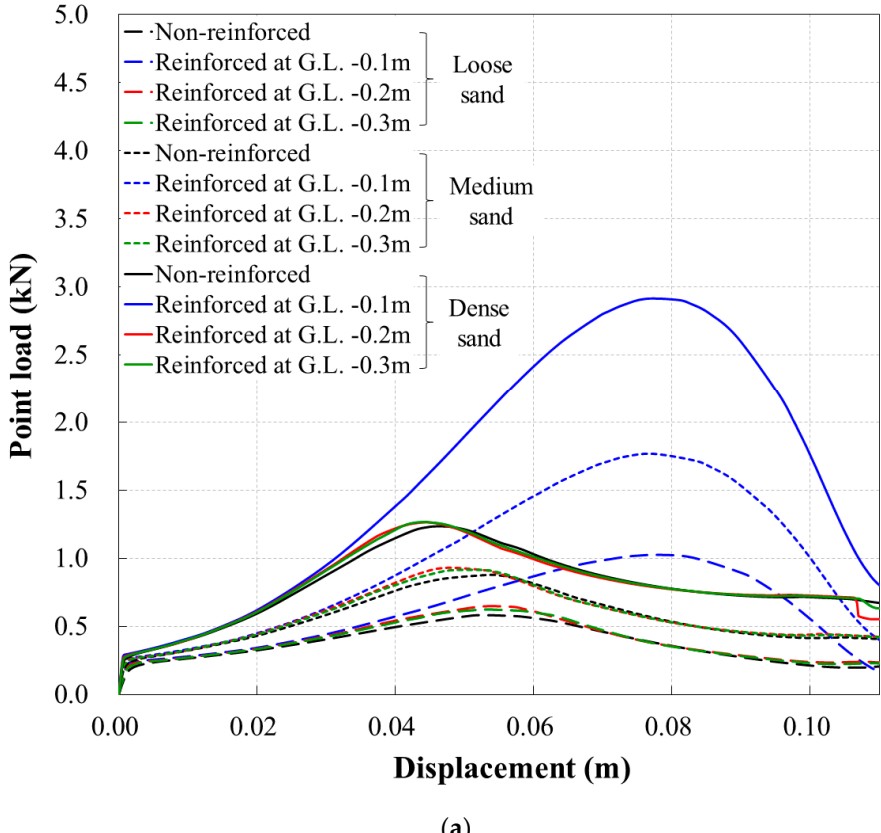

(**a**)

**Figure 9.** *Cont.*

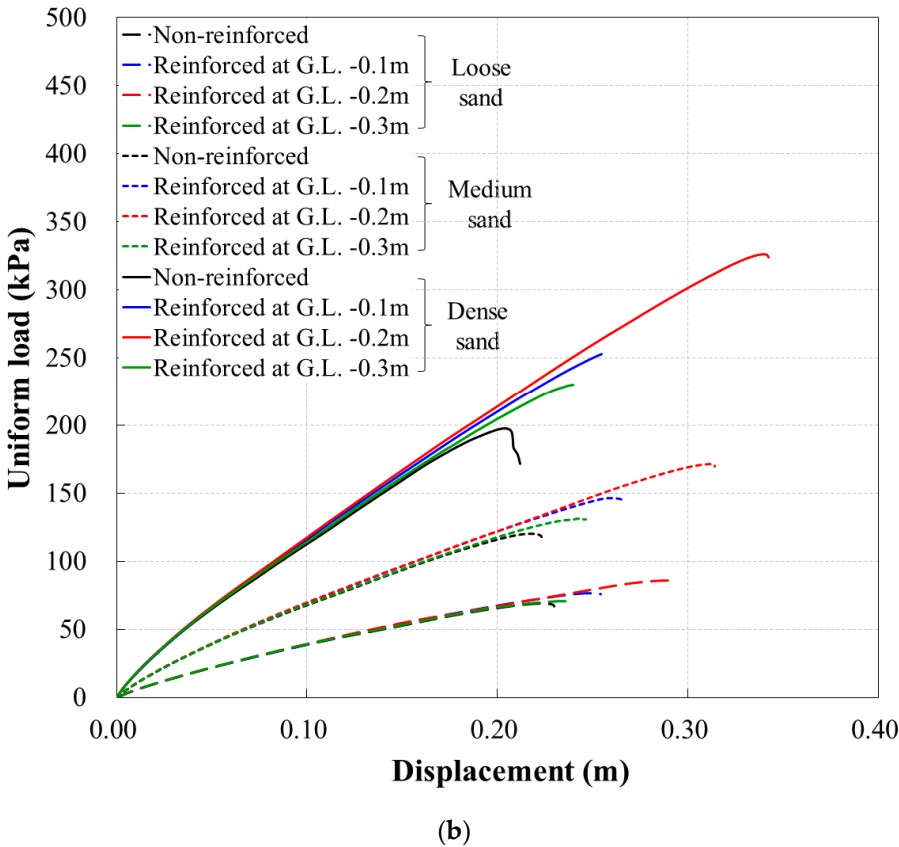

(**b**)

**Figure 10.** Relationship between the failure load and the displacement of the concrete mat according to the soil density and reinforced depth: (**a**) case of a point load; (**b**) case of a uniform load.

Figure 10b presents the results obtained when a uniform load was applied. For all analysis conditions, the displacement continued increasing as the load increased and the material then eventually failed. Specifically, the uniform load–displacement relationship was almost identical; however, the deformation rate did not increase after the failure load, unlike the case of the point load. The failure load almost increased by multiple times as the relative density increased. The order of the failure was from non-reinforced to G.L. −0.3 m, then to G.L. −0.1 m, and finally G.L. −0.2 m. The reason that the failure occurred earlier for the reinforced depth of G.L. −0.3 m than the other depths is assumed to be due to the failure that occurred in the soil above the concrete mat.

Based on the results above, a concrete mat can be used as a reinforcing method to reduce the stress generated below the concrete mat, according to the load conditions acting on the ground.

*4.3. Ratio of the Load Resistance Increase Caused by the Concrete Mat*

In order to quantitatively evaluate the reinforcement effect of the depth of a concrete mat, the ratio of the load resistance increase was analyzed by considering the installed depth of the concrete mat and the relative density of the soil per load type (Figure 11). The ratio of the load resistance increase was calculated by dividing the failure load of each depth and the relative density by the failure load of the non-reinforced case.

As shown in Figure 10, when the concrete mat was installed to GL −1.0 m and a point load was applied, the ratio of the load resistance increase was 75–130%, observed depending on the relative densities. Furthermore, the ratio of the load resistance increase was the highest in the soil with a higher relative density. Meanwhile, the increment ratio of load resistance significantly reduced when the depth increased, regardless of the relative density.

When a uniform load was applied, as presented in Figure 11b, the ratio of the load resistance increase was approximately 20–64% when it was installed at G.L. −0.2 m, depending on the relative

densities. The lowest ratio of the load resistance increase was observed when it was installed at G.L. −0.3 m.

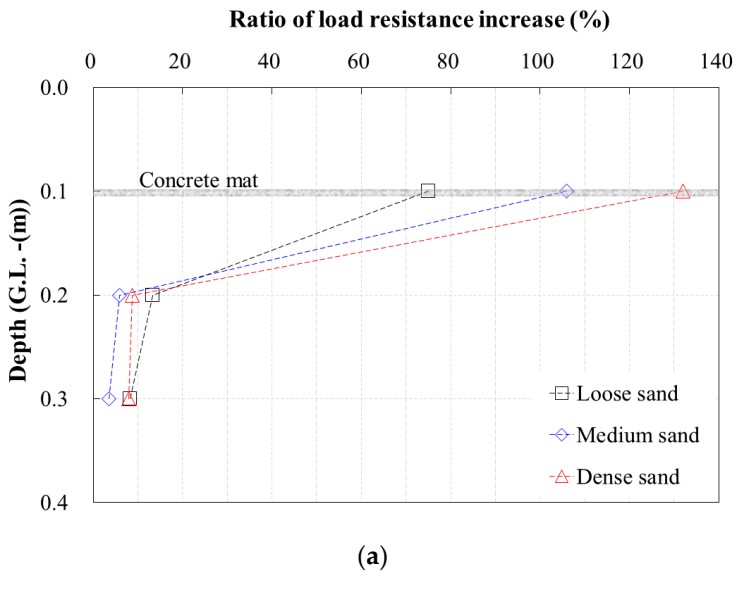

(**a**)

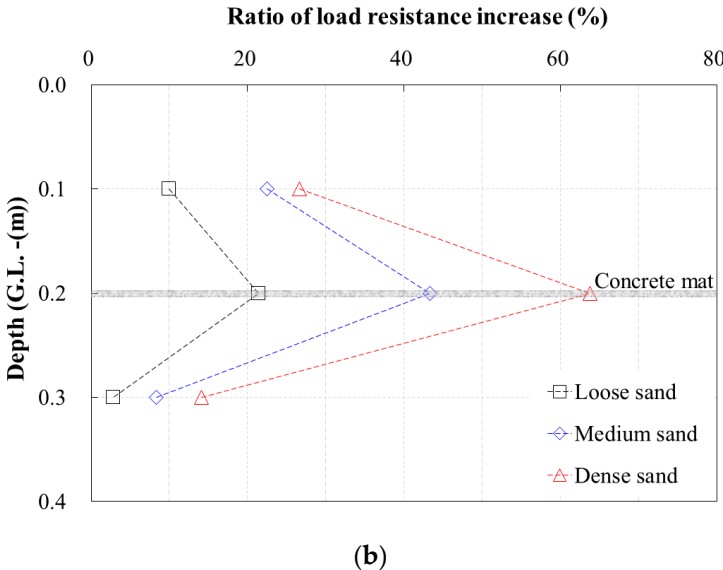

(**b**)

**Figure 11.** Ratio of the load resistance increase according to the soil density and reinforced depth: (**a**) case of a point load; (**b**) case of a uniform load.

## 5. Conclusions

In this study, a laboratory experiment and numerical analysis were performed using a concrete mat that was developed to prevent ground subsidence in order to evaluate the reinforcement effect. The results of this study are as follows:

1. As a result of the plate bearing test according to the reinforcement depth of the concrete mat, the stress was reduced by 28.5–30.9% below the reinforcement depth, confirming the reinforcement effect of the concrete mat;

2. In terms of performing a numerical analysis in which the various soil conditions and the reinforced depth are considered, the analysis model was verified against the experiment model. This study revealed that a lower stress reduction effect was observed in comparison to the experimental results. However, as the difference in the stress reduction effect was insignificant, while similar

tendencies were observed, the finite element analysis was sufficient to simulate the behavior and reinforcement effect of a concrete mat;

3.  Finite element analysis was performed while considering the relative density of the soil and the load type according to the reinforcement of a concrete mat. Compared with the non-reinforced soil, the ratio of the load resistance increase was 75–130% for a point load and 20–64% for a uniform load, respectively;

4.  As a result of finite element analysis, it was confirmed that the reinforcement effect according to the point load and uniform load was different, depending on the reinforcement depth. Therefore, the reinforcement effect of a concrete mat installed to prevent damage to a buried pipe can be maximized if the buried depth of the concrete mat is determined based on the type of load applied to the ground surface.

This study was conducted considering the simplified ground conditions. However, it is necessary to conduct a study on an analytical model considering various ground conditions (e.g., heterogeneous layered media), in order for the results of this study to produce a solution in engineering in the field.

**Author Contributions:** Conceptualization, J.P., Y.C., and G.H.; methodology, J.P.; validation, J.P. and Y.C.; formal analysis, G.H.; investigation, J.P., Y.C., and G.H.; resources, Y.C. and G.H.; writing—original draft preparation, G.H.; writing—review and editing, J.P. and Y.C.; visualization, G.H.; supervision, J.P.; funding acquisition, Y.C. All authors have read and agreed to the submitted version of the manuscript.

**Funding:** This research was funded by Korea Ministry of Environment: 2016000700001.

**Acknowledgments:** This research was supported by the Korea Environment Industry & Technology Institute (KEITI) through the Public Technology Program based on the Environmental Policy Project, funded by the Korea Ministry of Environment (MOE).

**Conflicts of Interest:** The authors declare no conflicts of interest.

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
