# Peer review of "Reinforcement Effect of a Concrete Mat to Prevent Ground Collapses Due to Buried Pipe Damage"

_applsci, doi:10.3390/app10165439_

Round 1

Reviewer 1 Report

The paper is in most of the paragraphs unreadable. Please reorganize the paper and also consider  extensive editing of English language and style is required in order to consider for review.

Author Response

Dear Reviewer.

Reviewer 2 Report

The study presented in this paper deals with the effect of concrete mat reinforcement on ground subsidence restoration.

The modification of stress distribution characteristics of underground induced by concrete mat reinforcement has been investigated by means of a laboratory test. In addition, a numerical analysis has been performed as a function of the different initial conditions, such as soil density and reinforced depth in order to estimate the range of the underground reinforcement of a concrete mat.

The paper is well written. The results are interesting and satisfactorily discussed.

The results could enlarge the knowledge about the behaviour of these type of reinforcement in the contest of ground subsidence.

I only suggest to move the sentence “The Atterberg limit test revealed that the soil sample does not have plasticity, and it was classified as poorly graded sand (SP) according to the Unified Soil Classification System.” (line 224-225 ) in the paragraph of Materials.

Author Response

Dear Reviewer.

Reviewer 3 Report

The authors investigated the effect of the concrete mat for failure in the vicinity of buried underground pipelines in the manuscript titled “Reinforcement Effect of a Concrete Mat to Prevent Ground Collapses Due to Buried Pipe Damage”. A numerical model is set up in commercially available software Abaqus and is validated against a laboratory-scale model. Then the calibrated model is used for sensitivity and variance analysis of controlling parameters such as density and depth of buried concrete mat. The manuscript read well and offers one of the solutions to a damaged pipeline scenario. However, the following points should be included before publication.

  1. Please add the least count of the LVDT sensors and the load measuring cell used in the experiment. Also, justify why one measurement is representative of a plane stress case.
  2. What is the length of the concrete mat used in the experiment? If I assume to the scale in figure 3, which is about 80-90% of the width of the box, a wall resistance will develop in the uniform granular media causing extra uplift pressure near the corners, which in turn will act as dense pillar points converting the assumption of uniform load on the sand to plate supported at corners. This causes a reduction in the pressure at the measurement point and one of the reasons for modelling mismatch and is shown in figure 8. See the figure below. The discrepancy could be check with placing more LVDT near the plate to see a uniform settlement or increasing/decreasing the box size or concrete mat.
  1. There is practically no information given about the material model and material parameters used for concrete and sand in the Abaqus model. A proper explanation should be provided to replicate the results and of practical use of the knowledge developed in the study.
  1. The numerical model is a powerful method; however, the problem studied here could also be modelled with settlement in a heterogeneous layered media. Such a solution is wide applicable by engineering in the field who require a fast and approximate solution. I would recommend adding this to improve the quality of the manuscript.
  2. Loose, medium and dense sand is used without giving any parameters of the material. I would suggest the authors add material properties and material models used in this study.
  3. The concrete used for the mat is somehow not talked. I would also suggest adding the property of concrete and strength parameters of the concrete used. The mix design will add an extra benefit to the audience and researchers.

For an explanation of relaxation zone at line 79, hardness for vice-a-verse of compactness “hardness or compactness around the excavation area…”. Hardness is a material property against abrasion or scratch. Remove this word as it confuses.

Author Response

Dear Reviewer.

Round 2

Reviewer 1 Report

Please find some comments in the pdf file

Author Response

Dear Reviewer

Reviewer 3 Report

Please check line 197, the strength of Concerte is wriiten wrong. Correct it before final draft. 

Author Response

Dear Reviewer
